# Highly active enzymes by automated combinatorial backbone assembly and sequence design

Gideon Lapidoth[1], Olga Khersonsky[1], Rosalie Lipsh[1], Orly Dym[2], Shira Albeck[2], Shelly Rogotner[2] & Sarel J. Fleishman[1]

Automated design of enzymes with wild-type-like catalytic properties has been a long-standing but elusive goal. Here, we present a general, automated method for enzyme design through combinatorial backbone assembly. Starting from a set of homologous yet structurally diverse enzyme structures, the method assembles new backbone combinations and uses Rosetta to optimize the amino acid sequence, while conserving key catalytic residues. We apply this method to two unrelated enzyme families with TIM-barrel folds, glycoside hydrolase 10 (GH10) xylanases and phosphotriesterase-like lactonases (PLLs), designing 43 and 34 proteins, respectively. Twenty-one GH10 and seven PLL designs are active, including designs derived from templates with <25% sequence identity. Moreover, four designs are as active as natural enzymes in these families. Atomic accuracy in a high-activity GH10 design is further confirmed by crystallographic analysis. Thus, combinatorial-backbone assembly and design may be used to generate stable, active, and structurally diverse enzymes with altered selectivity or activity.

[1] Department of Biomolecular Sciences, Weizmann Institute of Science, 7610001 Rehovot, Israel. [2] Israel Structural Proteomics Center, Weizmann Institute of Science, 7610001 Rehovot, Israel. These authors contributed equally: Gideon Lapidoth, Olga Khersonsky. Correspondence and requests for materials should be addressed to S.J.F. (email: sarel@weizmann.ac.il)

Enzymes can be grouped into families, members of which catalyze nearly identical chemical reactions, but exhibit vast differences in rates and substrate selectivities[1–3]. Conservation of chemical reactivity and diversity in substrate recognition are encoded in a modular architecture, wherein the residues actively taking part in catalysis are conserved in sequence and structure, typically including minute structural details. By contrast, structural elements outside the catalytic core vary substantially, including through insertion and deletion of large protein segments, to encode different substrate selectivities.

Enzymes belonging to the TIM-barrel fold, which is represented in five of the six top-level classes defined by the Enzyme Commission (EC)[3,4], are a prime example for this modularity. In each TIM-barrel family, eight parallel β-strands are arranged in a conserved and concentric barrel around the active-site pocket; the α-helices surround the strands and stabilize the pocket. By contrast to the atomic conservation of the catalytic residues in each family, the loops connecting the β-strands to the α-helices are highly variable in length, conformation, and sequence; substrate selectivity is largely encoded in these variable regions. Owing to this structural modularity, new substrate selectivities can evolve through gene recombination among homologous TIM barrels followed by insertion, deletion, and mutation; that is, as long as the scaffold's structural stability and the geometry of the core catalytic residues are maintained, the loop regions can vary substantially[5–7]. Indeed, more than 70 distinct sequence families in the Structural Classification of Proteins (SCOP) belong to the TIM-barrel fold[4,8], demonstrating how modularity has been exploited time and again by evolution. Structural modularity is a hallmark of other versatile enzyme classes, including, for instance, enzymes of the β-propeller, β-trefoil, Rossman, α/α-barrel, and α/β-hydrolase folds[9].

Modularity has also been exploited to optimize enzymes through laboratory evolution and structure-based recombination[10–12]. For instance, laboratory genetic recombination among naturally occurring enzymes through structurally conserved sites has generated enzymes with large variations in stability and specific activity[13–18].

Structure-based recombination has also been used to fuse TIM-barrel fragments and even fragments from unrelated folds, to generate new structures[19–22]. These and other structure-based and computational design studies[23–25] highlighted the structural adaptability of TIM barrels, but the resulting proteins were inactive, and in some cases, iterative laboratory evolution was employed, resulting in activities that were still several orders of magnitude lower than those of the wild type[18,22,26,27]. Furthermore, de novo enzyme design, whereby constellations of up to four catalytic residues are installed on natural scaffold proteins that do not exhibit the desired activity, targeted elementary reactions and has resulted in marginally stable proteins and catalytic efficiencies that were orders of magnitude lower than those of natural enzymes[28–30], similarly requiring iterative laboratory evolution to improve stability and rates and to obtain the designed active-site constellation[31–33]. Thus, automated design of stable and sophisticated enzymes exhibiting catalytic efficiencies that rival those of natural ones has been a long-standing though elusive goal[34–36].

Here, we demonstrate a path to automated design of stable and highly active enzymes. The design method is inspired by the evolution of new enzymes in nature through recombination, insertion, deletion, and mutation[37,38]. It starts by computationally segmenting all structures belonging to a modular enzyme family along structurally conserved sites and assembling the resulting modular fragments to generate a huge combinatorial diversity of backbones. Instead of using the natural sequences of the fragments, as in natural evolution or laboratory genetic recombination[39,40],

we next design the sequence of the entire protein (>300 amino acids) to maximize compatibility between the fragments and stabilize the active-site geometry. Each design step, therefore, introduces insertions and deletions as well as dozens of stabilizing mutations. Thus, although our method is inspired by natural evolutionary processes, each step is vastly more radical than individual recombination, insertion, deletion, and mutation events that occur in evolution, in which each event must be at least neutral in fitness or it is likely to be purged[37,41]. Despite having as many as 150 mutations from any natural enzyme, designed enzymes were stable, structurally accurate and highly active without requiring laboratory evolution. The results, therefore, provide proof-of-principle for fully automated design of stable, diverse, and highly active enzymes that catalyze complex reactions.

## Results

**The PLL and GH10 families**. To test the design method's generality, we targeted two structurally diverse and well-characterized TIM-barrel enzyme families exhibiting very different activity profiles: Phosphotriesterase-Like Lactonases (PLL) and Glycoside Hydrolase 10 (GH10) xylanases[42]. PLLs are a group of evolutionarily divergent enzymes[43] that possess a bi-metal center, which activates a water molecule for nucleophilic attack on the activated scissile bond of lactones (Fig. 1a). PLLs have potential applications in bacterial biofilm degradation and in the detoxification of organophosphates, including of highly toxic nerve agents[44–47]. GH10 enzymes hydrolyze the β-1,4 glycosidic bonds linking the xyloside units that comprise the backbone of the polysaccharide xylan, which is second only to cellulose in abundance in the plant cell wall[48]; xylanases are, therefore, essential for biomass degradation[49,50]. The GH10 catalytic core comprises two proximal and structurally fully conserved Glu sidechains, one acting as the nucleophile, which attacks the glycosidic bond, and the other as the protonating acid[51] (Fig. 1a). Furthermore, most PLLs are obligate homodimers (Supplementary Fig. 1), while GH10s are monomers, and the PLL family is highly diverse, including members with <25% pairwise sequence identity. Thus, the two enzyme families we targeted for design are unrelated in sequence, oligomeric state, active-site structure, and catalytic activity. In both families, the active-site pocket is complex and comprises positions from most β-α units (units 2–8 in GH10s and 1, 4, 5, 6, and 8 in PLLs, Fig. 1b). By contrast to previous enzyme-design studies, both reactions target biological rather than model substrates, and glycosidic-bond hydrolysis involves a high activation barrier, presenting an additional challenge.

In each family, some of the eight β-α backbone units are structurally highly diverse (Fig. 1c). Structural diversity presents opportunities for vastly increasing the number of potential backbones for design through combinatorial backbone assembly compared to the limited number of backbones observed in experimentally determined structures (114 and 154 for PLL and GH10, respectively). To estimate the potential for diversifying the backbones through combinatorial backbone assembly, we structurally clustered the eight β-α units in the GH10 family by 1 Å root-mean-square deviation (rmsd). We then computed the number of possible combinations, assuming that all β-α units could be recombined with all others, yielding a total of $10^{10}$ different backbones, exceeding the number of GH10 structures in the PDB (Protein Database) by eight orders of magnitude. We also reasoned that since all backbone fragments originate from natural enzymes, the likelihood of obtaining stable and functional enzymes from assembly is much higher than using naive scaffold libraries as in past enzyme design[28–30].

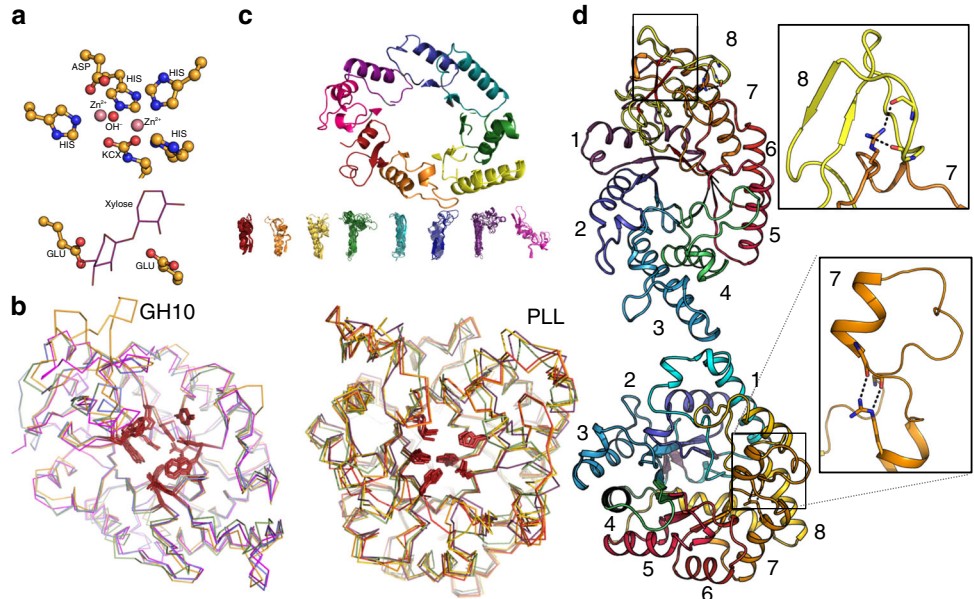

**Fig. 1** Combinatorial backbone assembly and design of enzymes. **a** Active-site constellations of the two targeted enzyme families: (top) PLLs comprising six chelating sidechains (including a carbamylated lysine, KCX), two metal ions, and a hydroxyl ion, and (bottom) GH10s comprising two catalytic Glu sidechains and the substrate. **b** Active-site pocket residues that are involved in substrate recognition (red sticks) are held fixed during design. **c** In combinatorial backbone assembly, homologous but structurally diverse enzymes are segmented along structurally conserved positions to produce exchangeable backbone fragments (bottom). The fragments are computationally recombined and sequence-optimized to generate new low-energy structures. **d** Irregular, long-range interactions, including buried charges, stabilize β-α backbone conformations. (top) GH10 β-α units 7 and 8 (PDB ID: 4PUE). (bottom) The long β-α loop (16 amino acids) of the *P. diminuta* phosphotriesterase (PDB ID: 2R1N)

**Combinatorial backbone assembly and design**. We segmented the structures in each family according to points of maximal structural conservation in the β-strands and extracted the backbone conformations of each segment for use in subsequent backbone assembly[52] (Fig. 1c). The choice of how to segment the backbone into β-α units is crucial for design success. Since our method samples backbone fragments independently of one another, each fragment must encode the most important stabilizing contacts; some stabilizing contacts, however, occur between adjacent fragments (Fig. 1d). To test the effects of different segmentation schemes on design success, we chose three segmentations for GH10 designs (Fig. 2): (1) A completely unbiased segmentation, in which each of the eight β-α units comprising the TIM-barrel were sampled independently of one another, maximally sampling backbone conformation space (design series xyl8); (2) A structure-based segmentation, comprising four backbone units, each of which forms stabilizing intrasegment contacts: β-α units 1, 2–4, 5–6, and 7–8 (series xyl4); and (3) another structure-based though discontinuous segmentation, where the least conformationally diverse segments, β-α units 1 and 5–6, formed one constant backbone segment and two other segments were formed by the variable units 2–4 and 7–8 (series xyl3). PLLs, by contrast to GH10s, are obligate homodimers[53,54]. We, therefore, sampled up to five fragments: one comprising the crucial homodimer interface formed by β-α units 1–3 and 8 and up to three other segments from units 4, 5, 6, and 7 (design series pll2, pll3, and pll4, which were assembled from 2, 3, or 4 PLL backbones, respectively). Thus, the computational design strategy is amenable to encoding a wide range of constraints inferred from experimental or structural analysis.

To assemble backbones and design new sequences, we generalized the Rosetta *AbDesign* method—originally developed to design new antibodies from backbone fragments of natural ones[52,55,56] (Supplementary Fig. 2, Supplementary Movie 1). For each of the segmentation schemes, we started from a random combination of backbone fragments. In each design step, *AbDesign* samples a single backbone fragment from the conformation database and designs the protein's amino acid sequence. Since the entire protein (>300 amino acids) needs to be designed to accommodate the large backbone changes introduced in each step, we used position-specific scoring matrices (PSSMs) to constrain amino acid choices at each position to identities that are commonly observed in a multiple-sequence alignment of natural family members. The PSSMs also focus design calculations on a sequence subspace that is more likely to include stable, folded, and active enzymes. Furthermore, the method does not model the enzyme-transition-state complex, which is often associated with modeling uncertainties and inaccuracies[33]. Instead, residues in direct contact with the bi-metal center in PLLs and the two catalytic Glu residues in GH10 as well as 11 additional residues directly involved in substrate binding, were not allowed to change sidechain conformation during design (Fig. 1b). Following sequence design, the new structure was accepted if it was lower in energy than the previous one, and higher-energy structures were accepted probabilistically. The designs were then ranked by Rosetta energy, clustered by backbone conformation to obtain conformationally unique structures (Supplementary Fig. 3) and subjected to the PROSS stability design algorithm in all regions outside the active-site pockets[57]. PROSS introduced dozens of mutations to each design (20 ± 6 and 36 ± 7 mutations in GH10 and PLL designs, respectively). Visual inspection indicated that the PROSS-designed mutations eliminated core cavities and improved surface polarity and were, therefore, likely to improve protein stability and expressibility.

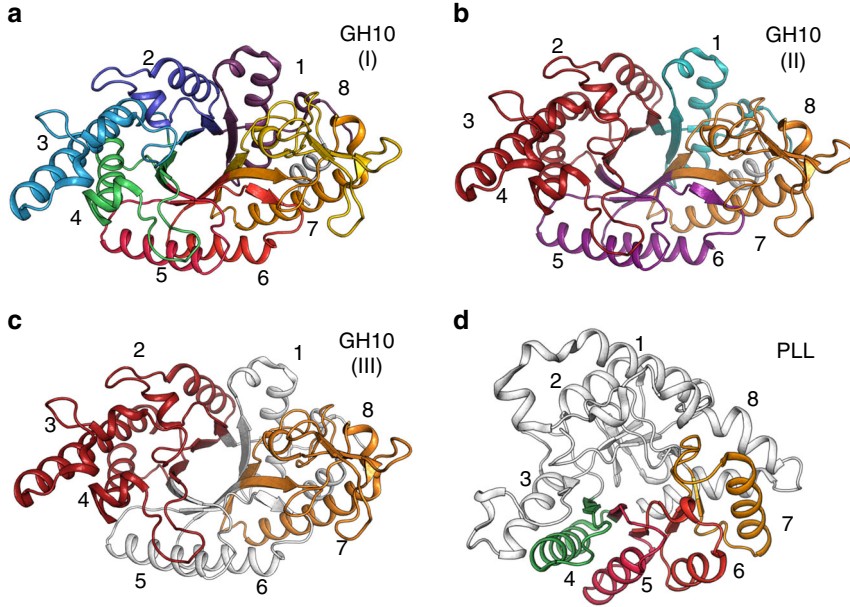

**Fig. 2** Different segmentation schemes used in combinatorial backbone assembly. **a** GH10(I): Each of eight β-α units were sampled independently, for maximal backbone diversity. **b** GH10(II): Segmenting β-α units 1, 2-4, 5-6, and 7-8 to preserve stabilizing interactions within each segment. **c** GH10(III): a discontinuous segmentation, in which the structurally conserved β-α units 1 and 5-6 formed one segment and two other segments were formed by units 2-4 and 7-8. **d** The homodimer interface in PLLs (β-α units 1-3 and 8, gray) was used as one backbone segment and units 4-7 were sampled independently

**Stable and highly active designs**. Synthetic genes encoding 34 PLL and 43 GH10 designs were fused C-terminally to maltose-binding protein (MBP), which served as a solubility and affinity-purification tag. The designs were then overexpressed in *E. coli* BL21 DE3 cells and purified using an amylose (PLL designs) or Ni-NTA (GH10 designs) column. All the designs expressed solubly, and >70% exhibited high expression yields (20–200 mg protein per liter of bacterial culture, Supplementary Fig. 4). Thus, despite as many as 150 mutations relative to any natural enzyme (Table 1, Supplementary Fig. 5), as MBP fusions, the designs did not require iterative rounds of in vitro evolution to optimize expressibility.

We initially screened each of the 43 GH10 designs for xylanase activity with a qualitative assay that measures the formation of reducing sugars released from natural beechwood xylan[58], finding that 20 of the 43 designs (46%) were active. We then selected the eight most active designs and two natural GH10 enzymes for quantitative kinetic analysis with the chromogenic substrate 4-nitrophenyl β-xylobioside (O-PNPX$_2$) (Table 1, Supplementary Tables 1 and 2 and Supplementary Fig. 6). The kinetic analysis revealed a wide range of catalytic efficiencies ($k_{cat}/K_M$), and encouragingly, the two most efficient designs, xyl3.1 and xyl3.2, exhibited rates within fivefold of natural GH10 family members (Fig. 3a, b), despite having >100 mutations relative to any natural GH10 enzyme (Table 1).

In many industrial applications, GH10 enzymes are subjected to high temperature and acidic pH. Some designs exhibited maximal activity at 45 °C, and some retained full activity even at 50 °C (Supplementary Fig. 7), with a pH optimum at 6–6.5 (Supplementary Fig. 8), similar to stability and activity profiles of natural GH10 enzymes. We thus concluded that the automated design method yielded several enzymes that were distant in sequence from any natural enzyme, yet showed similar catalytic and stability profiles to those observed in nature.

Natural PLLs exhibit a range of chemically related hydrolytic activities, including the hydrolysis of lactones, esters, and phosphotriesters. We initially tested the 34 designed PLLs with the artificial substrate 5-thiobutyl butyrolactone (TBBL)[59], finding seven active designs (Table 1). We subsequently measured the activity of these seven enzymes with a range of substrates: the natural aliphatic γ-nonanoic lactone, the ester p-nitrophenyl acetate, and the pesticide phosphotriester paraoxon (Supplementary Table 3; Fig. 3c, and Supplementary Figs. 9 and 10). PLL activity was tested with Co$^{2+}$ or Zn$^{2+}$, and in most cases much higher activity was observed with Co$^{2+}$, similar to previous reports[60]. Four PLL designs hydrolyzed the less activated aliphatic γ-nonanoic lactone, four exhibited esterase activity, and six hydrolyzed the phosphotriester paraoxon. Strikingly, the lactonase and esterase catalytic efficiencies of the four most active PLL designs were similar to those of natural PLLs (Table 1 and Fig. 3c, d). Indeed, design pll2.1 exhibited roughly twofold higher efficiency of TBBL hydrolysis and pll2.4 exhibited an order of magnitude higher efficiency in the hydrolysis of γ-nonanoic lactone and the pesticide paraoxon than the two natural enzymes. Hence, the designs exhibited features such as high catalytic efficiency and substrate promiscuity, while sampling sequence and conformation space widely (Supplementary Fig. 3). These designs can, therefore, be used as starting points for altering the selectivity profile or discovering new catalytic activities through active-site design or laboratory evolution.

Thermal stability is an essential property of enzymes in many biotechnological applications and low stability often constrains laboratory evolution of new activities[41]. Following overexpression of the active GH10 and PLL designs, we proteolytically cleaved the N-terminal MBP fusion, and subjected the enzymes to thermal denaturation, noting that all designs exhibited high apparent melting temperatures ($T_m$) in the range of 50–82 °C (Table 1 and Supplementary Figs. 11 and 12), comparable to the apparent $T_m$ of natural enzymes in these families, including enzymes from thermophiles.

**Atomic precision underlies high catalytic efficiency**. The active designs spanned five orders of magnitude in catalytic efficiency. For instance, the most active GH10 design xyl3.1 and the least

**Table 1 Parameters of active GH10 and PLL designs**

| Design/native enzyme | Mutations to nearest natural protein[a] | $T_m$, °C | $k_{cat}/K_M$, M$^{-1}$s$^{-1}$ [b] |
|---|---|---|---|
| xyl3.1 | 105 | 72 | 9417 ± 311 |
| xyl3.2 | 112 | 56 | 5060 ± 280 |
| xyl3.3 | 137 | 73 | 96 ± 10 |
| xyl4.1 | 141 | 50 | 36.3 ± 0.1 |
| xyl4.2 | 139 | 59 | 1.297 ± 0.004 |
| xyl8.1 | 159 | 61 | 156 ± 4 |
| xyl8.2 | 121 | 73 | 74 ± 4 |
| xyl8.3 | 130 | 57 | 0.61 ± 0.01 |
| Xylanase from *Geobacillus stearothermophilus* (PDB ID: 4PUD) | | 73 (78)[c] | 39,700 ± 3570 |
| pll2.1 | 80 | 74 (71) | 556,520 ± 13,810 (31,921 ± 1746) |
| pll2.2 | 54 | 79 (75) | 35,000 ± 780 (9944 ± 173) |
| pll2.3 | 75 | 82 (56, 67, 80) | 629 ± 15 (275.5 ± 4.9) |
| pll2.4 | 60 | 82 (68, 81) | 48,290 ± 1160 (70.6 ± 2.8) |
| pll2.5 | 67 | 60 (57) | 7.3 ± 0.3 |
| pll3.1 | 73 | 71 (68) | 667 ± 70 (4.02 ± 0.06) |
| pll4.1 | 85 | ND[d] (54, 74) | 21,060 ± 1100 (1655 ± 106) |
| PLL from *Sulfolobus solfataricus* (PDB ID: 2VC7) | | >100[80] | 210,000 ± 4447 (181,720 ± 3470) |
| PLL from *Geobacillus kaustophilus* (PDB ID: 4WVX) | | ND (55, 67, 81) | 77,370 ± 2106 (9356 ± 1182) |
| PTE from *Pseudomonas diminuta* (PDB ID: 1JGM)[e] | | 52 | 3460 |

The reported $k_{cat}/K_M$ values represent the means ± S.D. of at least two independent measurements
[a]Computed using BLASTP against the nonredundant database (nr). β-α unit composition of active designs is provided in Supplementary Table 5
[b]For PLL designs, $k_{cat}/K_M$ data are reported for 5-thiobutyl butyrolactone (TBBL) with CoSO$_4$, and in parentheses are given the data with ZnCl$_2$. Design pll2.5 was assayed only with ZnCl$_2$. For xylanase designs, the $k_{cat}/K_M$ data are reported for 4-nitrophenyl β-xylobioside (O-PNPX$_2$)
[c]$T_m$ determined by ThermoFluor; in parentheses—$T_m$ determined by NanoDSF. In several cases, multiple transitions were observed
[d]ND—no clear melting curve was obtained
[e]The inactivation temperature and the catalytic rate are given for the PTE-S5 variant from refs.[7,57]

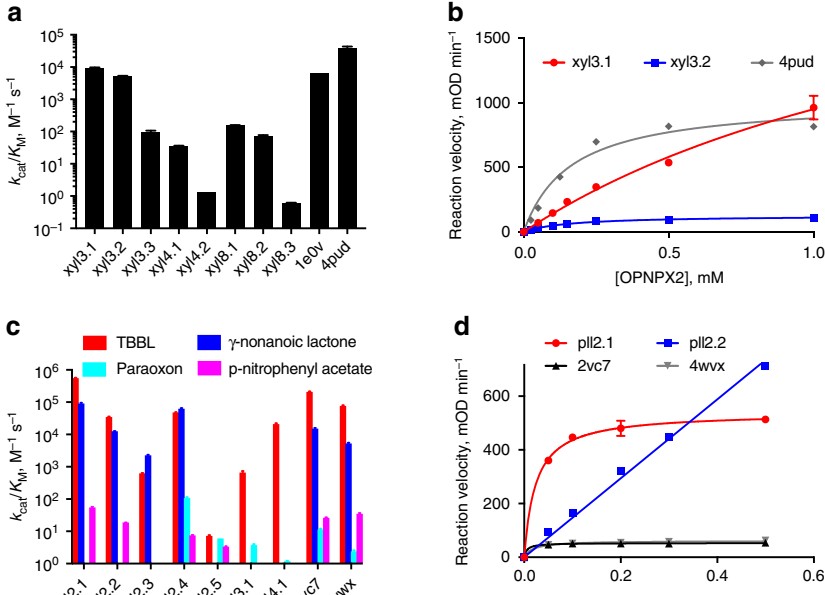

**Fig. 3** High catalytic efficiencies in GH10 and PLL designs. **a** Catalytic efficiency of GH10 designs with OPNPX$_2$ compared to natural GH10 enzymes (GH10s from *S. lividans* and *G. stearothermophilus*, PDB IDs 1E0V and 4PUD, respectively). Reaction velocities were normalized to 1 μM protein. **b** Michaelis-Menten curves of the most active GH10 designs relative to a natural GH10 (PDB ID: 4PUD). **c** Catalytic efficiency of PLL designs with various substrates compared to natural PLLs (from *S. solfataricus* and *G. kaustophilus*, PDB IDs 2VC7 and 4WVX, respectively). **d** Michaelis-Menten curves of the most active PLL designs and natural PLLs with TBBL. Reaction velocities were normalized to 0.1 μM protein. Data are the means ± standard deviation of duplicate reactions

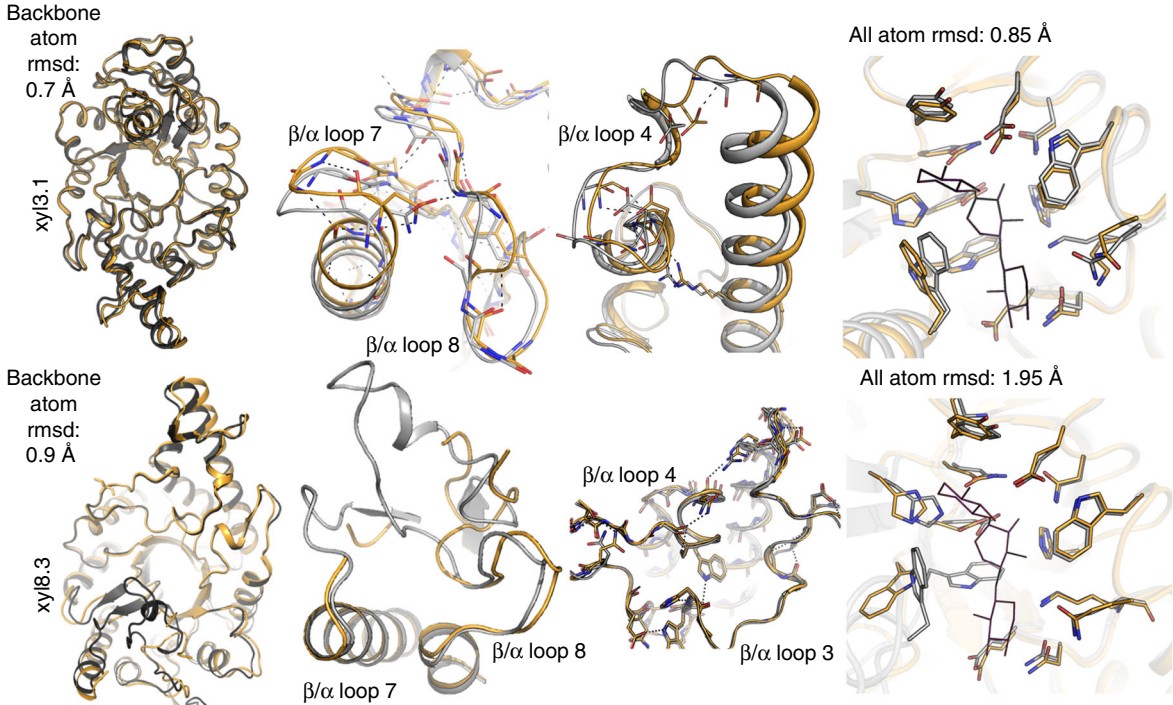

**Fig. 4** Design models vs. experimental structures. Shown are design models (gray) and crystal structures (gold) of high and low efficiency GH10 designs (xyl3.1 and xyl8.3, respectively). The designs were crystallized without substrate, and the coordinates of the substrate xylopentaose were extracted from PDB ID 4PUD (thin lines, right-hand side panels) after superimposing the designs' structures. The experimental structure of the high-activity design xyl3.1 is very similar to the design model throughout the backbone, as exemplified in the formation of long-range interactions between β-α loops 7 and 8 and within β-α segment 4 and in the orientations of all active-site residues. The experimental structure of the low-activity design xyl8.3 is similar to the model in most segments, for instance, β-α segments 3 and 4, and the two catalytic Glu residues, but loops in segments 7 and 8 failed to show substantial electron density

active one xyl8.3 exhibited $OPNPX_2$ hydrolysis efficiencies of 9417 and 0.61 $M^{-1}s^{-1}$, respectively. To understand what were the underlying structural reasons for this vast difference in efficiency, we determined their crystallographic structures (Fig. 4, Supplementary Fig. 13, and Supplementary Table 4). In both structures, the two catalytic Glu residues were positioned as in the design conception (<0.5 Å all-atom root-mean-square deviation (rmsd)). The high-activity design xyl3.1 was also atomically accurate throughout 13 active-site residues that form an intricate hydrogen-bond network surrounding the two catalytic Glu residues (<1 Å all-atom rmsd), and indeed, across the entire protein, with a total backbone rmsd of 0.7 Å. By contrast, the low-activity design xyl8.3 showed conformational changes in the residues that form the active-site hydrogen-bond network (2 Å all-atom rmsd). Furthermore, missing electron density in β-α loops 7 and 8 suggested that at least parts of the active-site pocket were mobile, although design accuracy was high throughout the segments of the design model and experimental structure that could be aligned (0.9 Å backbone rmsd). We, therefore, concluded that accurate positioning of the two catalytic Glu residues was crucial to obtain any level of activity, and that high levels of activity depended on atomic precision and preorganization in a large network of polar residues that surround the catalytic residues.

## Discussion
Combinatorial backbone assembly uses principles inferred from the evolution of enzyme families[38]: positions at the active-site and ones that are crucial for protein folding and stability are conserved, whereas large backbone and sequence changes, including insertions and deletions, generate vast structural diversity in regions that encode substrate selectivity. Resulting designs were

stable and some exhibited atomic accuracy and high catalytic efficiency and promiscuity even with respect to challenging non-activated substrates that are hydrolyzed by natural members of the respective enzyme families. By contrast, computational design of backbones at enzyme active sites has until now failed to show atomic accuracy, and stability and catalytic efficiencies were low[33,35,61,62], limiting the application of computational design to model reactions. It is also notable that expert-guided insertions and deletions at enzyme active sites are challenging and typically require rounds of trial-and-error and optimization, including in the PLLs that were the subject of our study[7]. These laborious iterative strategies can now be bypassed using combinatorial backbone assembly and design. The most active designs were based on fragments from only a few (two or three) different template enzymes, whereas combining fragments from four templates and more generally led to lower efficiency (Fig. 3). It, therefore, appears that future improvements to backbone assembly are needed to fully realize the potential of the design algorithm.

Our method exploits structure and sequence diversity in enzyme families to generate a large combinatorial diversity of backbones, followed by sequence design for stability. The ability demonstrated here to design enzymes exhibiting a large network of active-site residues at atomic accuracy and the resulting high catalytic efficiency greatly simplifies the goal of enzyme design: Instead of depending on accurate transition-state modeling, which, despite recent improvements, still suffers from uncertainty[33], conserving the natural active-site pocket suffices for wild-type like stability and catalytic efficiency in designs. Our study, therefore, demonstrates a fully automated path to design of structurally diverse enzymes, which catalyze complex reactions,

despite over 100 mutations from any natural enzyme. The resulting enzymes are stable, active, and highly diverse potential starting points for designing new substrate selectivities, providing an alternative to metagenomic screening and iterative in vitro evolution. An important future direction is to combine backbone fragments from non-homologous families[63], potentially extending the substrate spectrum beyond that observed within a target enzyme family. Thus, modular backbone assembly and design may provide a path to design of new biocatalysts.

## Methods

**A database of natural GH10 and PLL family enzymes.** 114 PLL and 154 GH10 structures were downloaded from the Pfam database[64,65]. The structures were segmented along structurally conserved points in the β-strands. For each segment, the mainchain dihedral angles ($\phi$, $\psi$, and $\omega$) and conformation-dependent PSSMs were computed using *AbDesign*[52].

**Segmentation positions.** PLLs were segmented along the following positions (for reference, position numbering is according to the PLL from *S. solfataricus*, PDB ID: 2VC7): β-α unit 4:135–169, 5:171–195, 6:197–217, 7:219–254. GH10 family enzymes were segmented along the following positions (for reference, position numbering is according to GH10 from *Thermoanaerobacterium saccharolyticum*, PDB ID: 3W24): β-α unit 1:20–44, 2:47–83, 3:86–142, 4:145–185, 5:188–218, 6:221–248, 7:251–288, 8:291–319.

**Constrained catalytic residues.** In PLLs, six metal chelating residues were constrained (for reference, position numbering is according to PLL from *S. solfataricus*, PDB ID: 2VC7): 22, 24, 137, 170, 199, and 256. In GH10s, the following active-site residues were constrained (for reference, position numbering is according to GH10 from *T. saccharolyticum*, PDB ID: 3W24): 52, 85, 89, 92, 145, 146, 187, 189, 221, 223, 251, 292, and 300.

**Combinatorial backbone assembly.** For each segmentation scheme, we generated a starting set of 3000 backbone conformations by randomly recombining fragments selected from the conformation databases. The sequence of each design was then optimized using RosettaDesign. From each starting design, 30 steps were taken, in each of which a backbone segment was chosen at random and replaced with a random fragment from the relevant conformation database. Following segment replacement, the sequence of the segment and every residue within 6 Å was designed, and iterations of sidechain packing, backbone and sidechain minimization were conducted to obtain low-energy sequences. The new structure was accepted relative to the previously accepted structure if it passed the Metropolis criterion with a gradually decreasing temperature (simulated annealing Monte Carlo).

**Structural clustering of designs.** All resulting designs were clustered using MaxCluster (http://www.sbg.bio.ic.ac.uk/maxcluster/).

**Stability design.** The clustered designs were subjected to the PROSS stability design algorithm[57], and for each starting design, one stabilized variant (PROSS design variant 6) was selected for experimental characterization.

**Rosetta energy function.** GH10 designs were computed using the Rosetta Talaris14 all-atom energy function[66], and PLL designs were computed using the more recent Rosetta energy function REF15[67]. Both energy functions are dominated by all-atom van der Waals packing, hydrogen bonding, electrostatics, and an implicit solvation model.

**Sequence identity analysis.** We calculated the sequence identity for each of the designed enzymes relative to the closest natural homolog using BLASTP[68] with the NCBI nonredundant (nr) database (ftp://ftp.ncbi.nlm.nih.gov/blast/db/).

**Design sequences.** Amino acid sequences of active designs are given in Supplementary Note 1.

**Materials.** Paraoxon, p-nitrophenyl acetate, γ-nonanoic lactone, 5,5-dithio-bis-(2-nitrobenzoic acid) (DTNB, Ellman's reagent), m-cresol, and beechwood xylan were purchased from Sigma-Aldrich. TBBL was kindly provided by the Tawfik laboratory[59]. 4-nitrophenyl β-xylobioside (PNPX$_2$) was purchased from Megazyme.

**Cloning.** Synthetic genes of designs and natural enzymes were codon optimized for efficient *E. coli* expression and custom synthesized as linear fragments by Twist Bioscience. The genes were amplified and cloned into the pETMBPH vector (containing an N-terminal 6-His-tag and MBP[69]) through the *Eco*RI and *Pst*I

restriction sites. The ligated DNA was transformed into *E. coli* BL21 DE3 cells, and DNA was extracted for Sanger sequencing to validate accuracy. The list of primers used for cloning is given in Supplementary Table 6.

**Protein expression and purification.** For small-scale expression, 2 ml of 2YT medium supplemented with 50 µg ml⁻¹ kanamycin (and 0.1 mM ZnCl$_2$ or CoSO$_4$ in case of PLLs) were inoculated with a single colony and grown at 37 °C for ~15 h. In all, 10 ml 2YT medium supplemented with 50 µg ml⁻¹ kanamycin (and 0.1 mM ZnCl$_2$ or CoSO$_4$ in case of PLLs) were inoculated with 0.2 ml overnight culture and grown at 37 °C to an OD$_{600}$ of ~0.6. Overexpression was induced with 0.2 mM IPTG, and the cultures were grown for ~24 h at 20 °C. After centrifugation and storage at −20 °C, the pellets were resuspended in lysis buffer and lysed by sonication.

PLL lysis buffer: 50 mM Tris (pH 8.0), 100 mM NaCl, 10 mM NaHCO$_3$, 0.1 mM ZnCl$_2$ or CoSO$_4$, benzonase and 0.1 mg ml⁻¹ lysozyme.

GH10 lysis buffer: 50 mM Tris (pH 6.8), 100 mM NaCl, benzonase and 0.1 mg ml⁻¹ lysozyme. The PLL proteins were bound to amylose resin (NEB), washed with 50 mM Tris pH 8.0 with 100 mM NaCl and 0.1 mM ZnCl$_2$ or CoSO$_4$, and eluted with wash buffer containing 10 mM maltose. The GH10 proteins were bound to Ni-NTA resin (Merck), washed with 50 mM Tris pH 6.8 with 100 mM NaCl and 20 mM imidazole, and eluted with wash buffer containing 250 mM imidazole. Elution fraction was used for SDS-PAGE gel and for initial activity measurements. For further analysis of active designs, the expression was repeated with 50 ml culture, and after purification, the proteins were dialyzed in wash buffer. For crystallization, the expression was performed with 500 ml culture, and after purification, the protein was digested with TEV protease to remove the MBP fusion tag (1:20 TEV, 1 mM DTT, 24–48h at room temperature (RT)). The MBP fusion was removed by binding to Ni-NTA resin, and the protein was purified by gel filtration (HiLoad 26/600 Superdex75 preparative grade column, GE). Protein concentration was estimated by OD$_{280}$ measurement, and protein expression levels were extrapolated to mg protein per liter culture.

**Preliminary xylanase screening.** Xylanase activity was determined qualitatively by measuring the reducing sugars released from xylan by the dinitrosalicylic acid (DNS) method[58]. A typical assay mixture consisted of 20 µl citrate buffer (500 mM, pH 6.0) added to 80 µl cell lysate. The reaction was started by adding 100 µl of 2% beechwood xylan suspended in DDW, and the reaction was continued for 20 min at 50 °C. The reaction was stopped by transferring the tubes to an ice-water bath. One-hundred microliters of the supernatant was then added to 150 µl DNS reagent, and the tubes were boiled for 10 min, after which the absorbance was measured at 540 nm. The read was compared to a blank sample (cell lysate expressing MBP), and active xylanase designs were taken for further examination.

**Kinetic measurements.** The kinetic measurements were performed with purified proteins (fused to MBP) in activity buffer (PLL: 25 °C, 50 mM Tris pH 8.0 with 100 mM NaCl, supplemented with 0.1 mM ZnCl$_2$ or CoSO$_4$, GH10: 37 °C, 50 mM Tris pH 6.5 with 150 mM NaCl). A range of enzyme concentrations was used, depending on the activity. The activity of PLLs was tested at 20–22 °C with TBBL[59], by coupling with DTNB and monitoring the absorbance at 412 nm), γ-nonanoic lactone (pH-sensitive assay in 2.5 mM bicine pH 8.3, by monitoring the absorbance of m-cresol indicator at 577 nm[70]), paraoxon and p-nitrophenyl acetate (monitoring the absorbance of the leaving group at 405 nm). The kinetic measurements were performed in 96-well plates (optical length –0.5 cm), and background hydrolysis rates were subtracted. The activity of GH10s was tested with O-PNPX$_2$ by monitoring the absorbance of the leaving group at 405 nm. No background hydrolysis was observed with O-PNPX$_2$. Specific activity of GH10s was also tested at a range of temperatures (25 °C, 37 °C, 45 °C, 50 °C) and at various pHs (citrate buffer: pH 5.0, 6.0, and 6.5, tris buffer: pH 7.0, 8.0, and 9.0).

**Determination of kinetic parameters.** Kinetic parameters were obtained by fitting the data to the Michaelis-Menten equation [$v_0 = k_{cat}[E]_0[S]_0/([S]_0 + K_M)$] using Prism 7. In cases where solubility limited substrate concentrations, data were fitted to the linear regime of the Michaelis-Menten model ($v_0 = [S]_0[E]_0 k_{cat}/K_M$) and $k_{cat}/K_M$ values were deduced from the slope. The reported values represent the means ± S.D. of at least two independent measurements.

**$T_m$ measurements.** $T_m$ measurements were performed after cleavage of the MBP tag from the designs. Two methods were used: ThermoFluor experiments using SYPRO Orange dye (Sigma-Aldrich) on a ViiA 7 real-time PCR machine, with temperature ramp from 25 °C to 100 °C at 0.05 °K s⁻¹ (ref. [71]), and nanoDSF experiments performed on Prometheus™ NT.Plex instrument (NanoTemper Technologies)[72]. In addition, residual activity of PLL designs was tested following 0.5 h incubation at various temperatures and cooling to RT.

**Structure determination and refinement.** Crystals of xyl3.1 and xyl8.3 were obtained using the hanging-drop vapor-diffusion method with a Mosquito robot (TTP LabTech). The crystals of xyl8.3 were grown from 8% PEG 3500 and 0.05 M Tri-sodium citrate dihydrate pH = 5.8. The crystals formed in the space group

$P4_12_12$, with one complex per asymmetric unit. A complete dataset to 2.1 Å resolution was collected at 100 °K on a single crystal on in-house RIGAKU RU-H3R X-ray. Crystals of xyl3.1 were grown from 0.5 M $(NH_4)_2H_2PO_4$ and 0.05 M sodium acetate pH=4.5. The crystals formed in the space group $H_3$, with one copy per asymmetric unit. A complete dataset to 1.85 Å resolution was collected at 100 °K on a single crystal on in-house RIGAKU RU-H3R X-ray.

Diffraction images of xyl3.1 and xyl8.3 crystals were indexed and integrated using the Mosflm program[73], and the integrated reflections were scaled using the SCALA program[74]. Structure factor amplitudes were calculated using TRUNCATE[75] from the CCP4 program suite. xyl3.1 and xyl8.3 structures were solved by molecular replacement with the program PHASER[76]. The models used to solve xyl8.3 and xyl3.1 structures were 3W25 and 3MMD, respectively.

All steps of atomic refinement of both structures were carried out with the CCP4/REFMAC5 program[77] and by Phenix refine[78]. The models were built into $2mF_{obs} – DF_{calc}$, and $mF_{obs} – DF_{calc}$ maps by using the COOT program[78,79]. Details of the refinement statistics of the xyl8.3 and xyl3.1 structures are described in Supplementary Table 4.

**Code availability**. All Rosetta design simulations used git version 2c0dc744fb56459daf220abc159f980b1809ecfe of the Rosetta biomolecular modeling software, which is freely available to academics at http://www.rosettacommons.org. The backbone conformation databases and PSSMs are distributed with the Rosetta release. RosettaScripts[64] and command lines are available in Supplementary Data 1–11.

**Data availability**. The coordinates of designs xyl8.3 and xyl3.1 are available from the RCSB Protein Data Bank (PDB IDs: 6FHE and 6FHF, respectively). Plasmids encoding the active designs are available from AddGene (IDs 107202–107217). Design protocols are available in the Supplementary Data 12–25. All other data supporting the findings of this study are available from the corresponding author upon reasonable request.

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

## Acknowledgements

We thank Melina Shamshoum, Lior Artzi, and Ed Bayer for help in establishing xylanase activity screens in our laboratory, and Nir London, Dan Tawfik, and members of the Fleishman lab for critical reading. The research was supported by a Starting Grant from the European Research Council (335439), the Israel Science Foundation through its Center of Excellence in Structural Cell Biology (1775/12) and its joint India-Israel Research Program (2281/15), and by a charitable donation from Sam Switzer and family.

## Author contributions

G.L., O.K., R.L., and S.J.F. designed the research; G.L. and O.K. contributed equally; G.L. and R.L. developed design methods and designed proteins; G.L. and O.K. performed biochemical experiments; O.D., S.A., and S.R. solved crystal structures; G.L., O.K., and S.J.F. wrote the manuscript with contributions from all authors; and S.J.F supervised the research. All the authors have given approval to the final version of the manuscript.

## Additional information

**Competing interests:** The authors declare no competing interests.

