## [Peer Review File · Nature Communications]

Reviewer #1 (Remarks to the Author):

This study by Fleishman and co-workers presents a method for enzyme diversification through combinatorial backbone assembly. This is an attempt at automated enzyme design with different properties, a very critical objective in protein engineering. By taking two structurally distinct templates, this work shows that the method can successfully assemble new backbone combinations and produce stable and active enzymes. Overall, I find the manuscript very clear and well written. The very rate of obtained active mutants from these designs is clearly an achievement and may be a key step in the automation of protein designs, as it may not require extensive following up with directed evolution approaches.

I have several comments on this work with the aim of helping the authors:

1) Last sentence of the abstract “...may be used to generate stable, active, and structurally diverse enzyme libraries with altered selectivity or activity”. Also applies to Conclusions.

While the authors have proven the two first points with their two examples (stable and active designs), I am more reserved in the structural diversity. The structural diversity observed in Fig. 4 is typical of engineering projects, comparing mutants' structures between themselves. It might be a terminology issue here, since I agree that the presented structures show levels of conformational / configurational diversity. However, shown structures are identical.

“altered selectivity or activity” . Data in Fig.3 suggests that some constructs have different selectivity, since they lost some catalytic, promiscuous activities. However, data does not support “altered activity” but rather “altered activity levels”.

I wonder if the authors tested whether some of these constructs might have gain functions?

2) P.7 1st paragraph” . “the designs did not require iterative rounds of in vitro evolution to optimize expressibility.

This appears to be an unfair statement, given the fact that the constructs were fused to MBP.

3) Were the catalytic parameters of the constructs determined with the MBP? Is it interfering with the measures at all?

4) P.9 1st paragraph: Is the active site all atom rmsd (2A rmsd) being compared to the remainder of the protein backbone rmsd? (0.9A)

It seems unreasonable to make a statement about accuracy outside the active site by comparing two very different measures.

Minor comments:

a) Did the authors look at the B-factors (normalized B) of these different structures? This could further illustrate the different conformational sampling of the designs, and fact that electronic density is not visible for a loop in the high efficiency design suggests that mobility is altered in these variants.

b) While the method was well described in the text, I think the manuscript would benefit of having a scheme explaining the different steps to the generation of the designs.

c) Table S3: There seem to be a very large difference on catalytic efficiency between the variants assayed in Co and Zn. Such a large difference is not observed for the wt. Is this suggesting that the design might have altered slightly the bi-metallic active site. Do the authors have any suggestions to

explain this?

d) Table S4: structure 6FHE: while the quality of the collected data seems excellent, the statistics for the refinement are very poor (e.g. Rfree ~29%, only 10 water molecules at 2.1Å resolution). Is there any explanation for this?

e) Table S4: please include CC1/2 statistics (Karplus and Diederichs, PMID: 26209821)

Reviewer #2 (Remarks to the Author):

The manuscript “Highly active enzymes by automated combinatorial backbone assembly and sequence design” submitted by the Fleishman group describes an *in silico* method for enzyme design. The algorithm, which is a straightforward combination of state-of-the-art methods, fulfills two tasks: These are the generation of a modified backbone by combining larger structural elements taken from known 3D structures and the subsequent computation of a proper protein sequence. The segments used for backbone assembly by means of Rosetta are from homologous enzymes that share the same fold and function. The generalized Rosetta protocol used by the authors generates protein sequences as well. Here, these sequences are subsequently subjected to the PROSS server, which introduces stabilizing mutations.

The authors generated 77 sequences, the corresponding proteins were expressed in *Escherichia coli* and in contrast to many other design experiments, all were soluble and approximately 36% exhibited enzyme activity comparable to wild type enzymes. Based on these encouraging findings, the authors recommend their method as a general tool for the design of enzymes with altered selectivity or activity.

The manuscript is well written and the methods and the results are clearly presented. However, it is unclear whether the authors are too optimistic with respect to their conclusions. First, both enzymes under study possess a TIM-barrel fold, which is known to tolerate a large number of modifications. Thus, the success of the method in designing enzymes with less robust folds is unknown. Second, the side-chain orientation of crucial residues remains fixed during the design process. As the authors demonstrate, this is key to success, but also a limitation, making it difficult to extend the substrate spectrum or the design of non-wild type enzyme functions. To put it another way, the authors have developed a method that alters with great success structure and sequence but conserves stability and function of the wild type enzymes. The authors should rephrase their text and mention these limitations more clearly. Nevertheless, compared to alternatives, this is a big achievement.

The authors use different libraries of structural elements and in both enzyme designs, success depended on the size and the number of the backbone fragments chosen for the initial assembly. Comparing the results, I deduced that less complex designs were more successful. Thus, I recommend to mention this tendency. A further detail important for all users of the Rosetta suite is the number of mutations introduced by the PROSS server. The authors should list these values; then readers can decide on the relevance of this additional optimization step.

Just to improve consistency and readability, as I have noticed some typos:

Usually, a full genus name is given on the first use and abbreviated only after the first use. Legend to Table 1: (a) The meaning of “Blade composition” is unclear to me. Legend to Figure 4: “The ... design xyl3.1 exhibits high accuracy...” is unclear to me. I recommend proofreading the bibliography to correct formatting errors and typos e.g. those concerning the author names Höcker and Röthlisberger. The correct typesetting is “Position-Specific Scoring Matrices”. To name PDB data sets, the abbreviations PDB entry, PDB ID, and PDB were used. The formatting of the Supplement can be improved and should be standardized.

Reviewer #3 (Remarks to the Author):

One of the key challenges in protein design - whether the target is an antibody, an enzyme or a therapeutic - is effective sampling of backbone degrees of freedom. Significant backbone rearrangements in design are a way to facilitate diversity - at least at the structural level. This work on enzyme design takes a smart approach to generating backbone-diverse libraries by carefully choosing break points in feedstock sets of structures and recombining them through the powerful tools of the ROSETTA platform.

The success rate of the approach is high, approaching 50%. The functional potential of the library is also high with candidates approaching WT-activity and many having significant thermostability. This makes the library very attractive as a starting point for in vitro evolution.

The manuscript is technically sound, clearly written - and is likely to have a significant impact on the field of protein engineering and design. I have no issues with its publication in its present form.

Reviewer #1:

Last sentence of the abstract “...may be used to generate stable, active, and structurally diverse enzyme libraries with altered selectivity or activity”. Also applies to Conclusions. While the authors have proven the two first points with their two examples (stable and active designs), I am more reserved in the structural diversity. The structural diversity observed in Fig. 4 is typical of engineering projects, comparing mutants’ structures between themselves. It might be a terminology issue here, since I agree that the presented structures show levels of conformational / configurational diversity. However, shown structures are identical.

Right. We have added Fig. S3 to show the level of structural diversity among the active designs, and refer to this in pg. 8.

“altered selectivity or activity” . Data in Fig.3 suggests that some constructs have different selectivity, since they lost some catalytic, promiscuous activities. However, data does not support “altered activity” but rather “altered activity levels”.

I wonder if the authors tested whether some of these constructs might have gain functions?

To improve accuracy, we have revised to “altered activity levels” in the Abstract, as the Reviewer suggested. The designs do not show gain of function relative to known enzymes from the respective natural families, as far as our assays indicated.

P.7 1st paragraph” . “the designs did not require iterative rounds of in vitro evolution to optimize expressibility.

This appears to be an unfair statement, given the fact that the constructs were fused to MBP.

MBP fusion is indeed important for solubility, and is very commonly used in the engineering literature on these enzymes, including in the *in vitro* evolution field (for example, see PMID:17105187, PMID:22809311). To avoid any confusion, we have clarified the statement pointed by the reviewer to indicate that “as MBP fusions”, the designs did not require further optimisation.

Were the catalytic parameters of the constructs determined with the MBP? Is it interfering with the measures at all?

We now explain that catalytic activities of both the designs and the natural enzymes were assayed as MBP fusions. Enzymes from the PLL family are routinely assayed with MBP (for example, see PMID:17105187, PMID:22809311), and diffusion-limited activity was obtained for PTE in our hands, so MBP is certainly not harming the activity. We note that the MBP-fusion construct contains a 21-aa spacer between MBP and the enzyme, and MBP is therefore unlikely to impact activity.

P.9 1st paragraph: Is the active site all atom rmsd (2A rmsd) being compared to the remainder of the protein backbone rmsd? (0.9A)

It seems unreasonable to make a statement about accuracy outside the active site by comparing two very different measures

In this paragraph, we did not intend to compare rmsds within and outside the active-site pocket, we were merely indicating that the active site pocket in the low-efficiency design was not accurate (in all-atom rmsd), whereas the remainder of the scaffold was quite accurate. We clarified this paragraph.

Did the authors look at the B-factors (normalized B) of these different structures? This could further illustrate the different conformational sampling of the designs, and fact that electronic density is not visible for a loop in the high efficiency design suggests that mobility is altered in these variants.

This is a very good point. We went back to the structures, and although the absolute B-factors imply higher mobility in the active-site pocket of the less efficient design, the normalised B-factors do not. This is because the average B-factors in the low-efficiency design is much higher than in the efficient design, so, unfortunately, it does not seem that we can convincingly make this argument.

While the method was well described in the text, I think the manuscript would benefit of having a scheme explaining the different steps to the generation of the designs.

Right. We have added Supplemental Fig. 2 and a movie (Supplemental Movie) to explain the design method. These are cited on pg. 6.

Table S3: There seem to be a very large difference on catalytic efficiency between the variants assayed in Co and Zn. Such a large difference is not observed for the wt. Is this suggesting that the design might have altered slightly the bi-metallic active site. Do the authors have any suggestions to explain this?

Yes, there are large differences due to metal composition, but such changes are also observed in natural PLLs (4wvx exhibits eightfold differences in activity with different metals). We now note that these differences were also observed in a previous report (pg. 8). The atomic differences among the metals could directly contribute to this observed change, and the designs may have indeed changed the bimetal center to some extent. Without high-resolution experimental structures with different metals, however, we do not think that we can make a convincing argument.

Table S4: structure 6FHE: while the quality of the collected data seems excellent, the statistics for the refinement are very poor (e.g. Rfree ~29%, only 10water molecules at 2.1A resolution). Is there any explanation for this?

Due to the reviewer's concern, we tried to refine the structure further by adding water molecules, but did not see an appreciable improvement in refinement statistics. We added approximately 40 water molecules, but the R-free was still high (28.6%), with no appreciable effect on R-factor. We also ran PDBREDO but the R-free was not improved. Based on the Xtriage program from PHENIX, the data look good except for a comment indicating that data are "moderately anisotropic", which may contribute to the high R-free. Also the average B-factor is quite high for this resolution which might be due to the high structural mobility in this structure, which could in turn result in higher R-free. Given that the refinement statistics did not improve through the addition of water molecules, we decided not to add them to the deposited structure.

Table S4: please include CC1/2 statistics (Karplus and Diederichs, PMID: 26209821)

Done.

Reviewer #2

The manuscript is well written and the methods and the results are clearly presented. However, it is unclear whether the authors are too optimistic with respect to their conclusions. First, both enzymes under study possess a TIM-barrel fold, which is known to tolerate a large number of modifications. Thus, the success of the method in designing enzymes with less robust folds is unknown. Second, the side-chain orientation of crucial residues remains fixed during the design process. As the authors demonstrate, this is key to success, but also a limitation, making it difficult to extend the substrate spectrum or the design of non-wild type enzyme functions. To put it another way, the authors have developed a method that alters with great success structure and sequence but conserves stability and function of the wild type enzymes. The authors should rephrase their text and mention these limitations more clearly. Nevertheless, compared to alternatives, this is a big achievement.

Regarding the first point, the method is indeed limited only to modular folds, as described throughout the text, and we also mention some specific modular folds in the Introduction. To avoid any confusion, we have now added to the abstract that the two targets belong to the TIM-barrel fold. Still, while TIM barrels are tolerant of modifications, much of the literature on TIM barrel assembly from fragments did not result in any activity, let alone wild type levels of activity as observed in our study. The PLL family that is one of the targets of our study was also studied by *in vitro* evolution to change the backbone of a single β - α unit in the barrel, but that study showed that obtaining functional protein after even such a relatively minor modification comes at great difficulty (PMID: 22809311). Regarding the second point, yes, this design strategy can change selectivity but not extend the substrate spectrum. We now explain that we change activity **levels** rather than activities (Abstract), and in the conclusions, explicitly mention that the more challenging goal of extending the substrate spectrum is still to be achieved (though it may be made possible by recombining fragments from different families) (pp. 10-11).

The authors use different libraries of structural elements and in both enzyme designs, success depended on the size and the number of the backbone fragments chosen for the initial assembly. Comparing the results, I deduced that less complex designs were more successful. Thus, I recommend to mention this tendency.

True. We have added a statement that the less complex combinations are more successful, suggesting room for improvement in the design method (pg. 10).

A further detail important for all users of the Rosetta suite is the number of mutations introduced by the PROSS server. The authors should list these values; then readers can decide on the relevance of this additional optimization step.

Right. PROSS was very important in this workflow. We now mention the numbers (dozens of mutations introduced by PROSS) and also that before introducing the PROSS stabilisation step, visual inspection suggested that the designs contained many structural defects (pg. 7).

Just to improve consistency and readability, as I have noticed some typos:

Usually, a full genus name is given on the first use and abbreviated only after the first use.

Legend to Table 1: (a) The meaning of "Blade composition" is unclear to me. Legend to Figure 4: "The ... design xyl3.1 exhibits high accuracy..." is unclear to me. I recommend proofreading the bibliography to correct formatting errors and typos e.g. those concerning the author names Höcker and Röthlisberger. The correct typesetting is "Position-Specific Scoring Matrices". To name PDB data sets, the abbreviations PDB entry, PDB ID, and PDB were used. The formatting of the Supplement can be improved and should be standardized.

Thank you! All done.

Reviewer #3

One of the key challenges in protein design - whether the target is an antibody, an enzyme or a therapeutic - is effective sampling of backbone degrees of freedom. Significant backbone rearrangements in design are a way to facilitate diversity - at least at the structural level. This work on enzyme design takes a smart approach to generating backbone-diverse libraries by carefully choosing break points in feedstock sets of structures and recombining them through the powerful tools of the ROSETTA platform.

The success rate of the approach is high, approaching 50%. The functional potential of the library is also high with candidates approaching WT-activity and many having significant thermostability. This makes the library very attractive as a starting point for in vitro evolution.

The manuscript is technically sound, clearly written - and is likely to have a significant impact on the field of protein engineering and design. I have no issues with its publication in its present form.

We thank all the reviewers for their support and constructive comments.